# Gold Nanoparticles as Effective ion Traps in Poly(dimethylsiloxane) Cross-Linked by Metal-Ligand Coordination

**DOI:** 10.3390/molecules27113579

**Published:** 2022-06-02

**Authors:** Angelika Wrzesińska, Emilia Tomaszewska, Katarzyna Ranoszek-Soliwoda, Izabela Bobowska, Jarosław Grobelny, Jacek Ulański, Aleksandra Wypych-Puszkarz

**Affiliations:** 1Department of Molecular Physics, Faculty of Chemistry, Lodz University of Technology, Zeromskiego 116, 90-924 Lodz, Poland; izabela.bobowska@p.lodz.pl (I.B.); jacek.ulanski@p.lodz.pl (J.U.); 2Department of Materials Technology and Chemistry, Faculty of Chemistry, University of Lodz, Pomorska 163, 90-236 Lodz, Poland; emilia.tomaszewska@chemia.uni.lodz.pl (E.T.); katarzyna.soliwoda@chemia.uni.lodz.pl (K.R.-S.); jaroslaw.grobelny@chemia.uni.lodz.pl (J.G.)

**Keywords:** gold nanoparticles, metal-ligand coordination, poly(dimethylsiloxane), broadband dielectric spectroscopy, ionic charge carrier trapping

## Abstract

At this time, the development of advanced elastic dielectric materials for use in organic devices, particularly in organic field-effect transistors, is of considerable interest to the scientific community. In the present work, flexible poly(dimethylsiloxane) (PDMS) specimens cross-linked by means of ZnCl_2_-bipyridine coordination with an addition of 0.001 wt. %, 0.0025 wt. %, 0.005 wt. %, 0.04 wt. %, 0.2 wt. %, and 0.4 wt. % of gold nanoparticles (AuNPs) were prepared in order to understand the effect of AuNPs on the electrical properties of the composite materials formed. The broadband dielectric spectroscopy measurements revealed one order of magnitude decrease in loss tangent, compared to the coordinated system, upon an introduction of 0.001 wt. % of AuNPs into the polymeric matrix. An introduction of AuNPs causes damping of conductivity within the low-temperature range investigated. These effects can be explained as a result of trapping the Cl^−^ counter ions by the nanoparticles. The study has shown that even a very low concentration of AuNPs (0.001 wt. %) still brings about effective trapping of Cl^−^ counter anions, therefore improving the dielectric properties of the investigated systems. The modification proposed reveals new perspectives for using AuNPs in polymers cross-linked by metal-ligand coordination systems.

## 1. Introduction

Flexible and stretchable electronics constitute today an emerging technology expected to find multiple applications in the construction of novel optoelectronic devices such as foldable displays, wearable photovoltaic cells, extensible sensors, medical devices, etc. Furthermore, these devices might be easily manufactured at relatively low temperatures and by simple and cost-effective printing processes, thus leading to a substantial cost reduction [1]. The development of modern electronic circuitry demands a range of new materials having different functionalities, beginning with substrates and semiconductors through gate dielectrics and up to the conductive materials for electrodes [2]. All these components should fulfill specific electrical requirements while demonstrating mechanical properties suitable for stretchable and flexible applications. Combining stretchability with a desired electronic functionality constitutes a real challenge in the field of modern materials engineering.

A gate dielectric is an example of a material that has been intensively optimized with regard to its composition and processing methods in order to meet specific requirements for thin-film transistors—vital components of each electronic circuit [1,3]. Dielectric permittivity (often represented by relative permittivity k = ε/ε_0_) and dielectric loss are two significant parameters that characterize the gate dielectric. For optimum working parameters of a thin-film transistor, low dielectric loss (<0.01) and high k values are required from an ideal candidate for a gate dielectric. The high k-value decreases the operating voltage of the transistor. However, in the case of very high k values, bias stress behavior is observed that deteriorates the working parameters of the transistor. Both these factors are dependent on the internal structure of the gate dielectric and on its homogeneity in particular. Thus, a precise design of the material’s molecular composition allows elaborated tailoring of its electrical parameters.

Polymers and polymer-based hybrid materials are most suitable for flexible and stretchable electronics. So far, many different approaches have been proposed in order to obtain an optimum gate dielectric composition. Poly(vinylidene fluoride) is a commonly used polymer with an elevated value of permittivity (k). However, its crystalline nature brings about high surface roughness, thus introducing defects into the semiconductor layer, as well as non-uniformity of the electric field. This, in turn, causes location/trapping of the charge carriers and, as a result, a reduction in their mobility. To solve this problem, copolymers characterized with smother morphology such as poly(vinylidenefluoride-co-trifluoroethylene) were proposed. However, at the same time, they exhibit a higher value of dielectric permittivity, thus resulting in a deterioration of the transistor’s working parameters. To solve this problem, Guo et al. [4] proposed a bilayer structure of high-k/low-k poly(vinylidene fluoride-trifluoroethylenechlorofloroethylene/poly(methylmethacrylate) copolymer. Such a modification made it possible to manufacture low-voltage organic thin-film transistors with suitable bending stability. Other works describe the application as a gate dielectric of a variety of composite materials: polyvinyl alcohol/SiO_2_ [5], ZrTiHfO_2_-PVP [6], poly(vinyl alcohol)/2D TiO_2_ nanosheets [7], as well as block copolymer elastomers [8,9].

A significant fraction of the ongoing works concerns polydimethylsiloxane (PDMS), often used in flexible electronics as a substrate material [10] (due to its excellent mechanical properties), as well as gate dielectric [11] (despite its low dielectric permittivity). Different ways of PDMS modification were reported in the literature, which allowed for tunning permittivity [12] or adding new functionalities such as self-healing properties [13]. Rajitha and Dash proposed a PDMS/reduced graphene oxide composite that preserved optical transparency while increasing the permittivity up to 38 [14]. The addition of ceramic such as BaTiO_3_ [15], SrTiO_3_ [16], as well as metallic [17] nanoparticles also caused an increase in permittivity of PDMS-based composites. Moreover, on a nanometric scale, they exhibited unusual optical and electrical properties due to quantum confinement. It was observed that the addition of metal nanoparticles into a polymer dielectric matrix inhibits conduction loss due to the Coulomb blockade effect [18]. However, for low loading levels, they provide only a slight enhancement of dielectric properties [19]. In the case of higher loading levels, on the other hand, conductive elastomers might be obtained [20]. Uddin et al. [21] described a convenient biobased synthesis of a polyvinyl alcohol composite with Ag nanoparticles. The authors observed a remarkable enhancement of permittivity, reaching up to 900, as well as dielectric loss equal to 0.14. It was Feng et al., in turn, who proposed a composite of silver and nickel nanoparticles embedded in the PDMS matrix—a system characterized by very low dielectric loss—0.009 for 1 MHz at room temperature and a maximum dielectric permittivity of 35 [17].

Metal nanoparticle loading is not the only parameter affecting the electrical behavior of a composite; other chemical reagents and processing methods also have a significant contribution to the dielectric response. Despite a broad range of reported copolymers and composites, there is still a lack of systems fulfilling all the requirements of flexible and stretchable electronics. Therefore, there is still a strong demand for new functional materials.

The present work describes, for the first time, how the addition of gold nanoparticles (AuNPs) affects the electrical properties of PDMS cross-linked with a metal-bipyridine (bpy) coordination. For that purpose, a polymer system cross-linked with zinc chloride (ZnCl_2_) salt was selected as the best system within those investigated in our previous studies [22,23]. The reported bpyPDMS-ZnCl_2_ coordination exhibits the highest increase in both the real part of dielectric permittivity (ε′) and direct current (DC) conductivity, observed under low-frequency/high-temperature conditions. The described combination was used as a model one to study the influence of AuNPs on the electrical properties of the previously tested systems.

## 2. Materials

### 2.1. Synthesis, Functionalization, and Characterization of the Nanoparticles

Materials: gold (III) chloride hydrate (Sigma-Aldrich, ≥49%), sodium borohydride (NaBH_4_, purity ≥ 96%, Sigma-Aldrich), thiol-terminated polystyrene (PSSH, Sigma-Aldrich, Mn = 11.000) were used as received. For an aqueous colloid preparation, deionized water obtained from the Deionizer Millipore Simplicity UV system was used (the specific resistivity of water was equal to 18.2 MΩ⋅cm).

At first, AuNPs were synthesized in water via the chemical reduction method. Briefly, chloroauric acid water solution (3.807 g, 0.136 wt. %) and water (25.179 g) were added to a flat bottom flask and mixed vigorously for 5 min at room temperature. Next, sodium borohydride (1.015 mL, 0.5 wt. %) was added, and the solution was mixed for an additional 1 h. The final concentration of AuNPs in an aqueous colloid was equal to 100 ppm. The hydrodynamic size of the nanoparticles and their colloidal stability in water were investigated with the help of Dynamic Light Scattering (DLS). The shape, size, and size distribution of the metallic core were analyzed using High-Resolution Scanning Transmission Electron Microscopy (HR-STEM).

The functionalization of AuNPs with PSSH ligand was performed via a phase transfer process [24,25,26,27] from water to toluene according to the procedure described in our previous work [27]. The functionalization procedure was as follows: to the 30.0 g of aqueous AuNPs colloid, 15.0 g of acetone, and 15.0 g of toluene PSSH solution (0.113%) were added, with the solution being vigorously mixed for 15 min and followed by phase separation for 1 h. Acetone was used to reduce the surface tension between the water and toluene phases and to increase the miscibility of both phases. The weight ratio of aqueous colloid to acetone and toluene was equal to 2: 1: 1, respectively. The surface coverage of AuNPs with PSSH was equal 5 ligand molecules per 1 nm^2^ of the nanoparticle surface. The efficiency of AuNPs phase transfer was observed visually as the color of the aqueous phase changed from red to yellow, and the organic phase became red. Next, the organic colloid was evaporated under reduced pressure. The process was carried out in two steps, first at 100 mbar for about 30 min to remove acetone, then at 15 mbar for about 1.5 h to concentrate the colloid. The final AuNPs-PSSH concentration in toluene was equal to 2500 ppm. The colloidal stability, hydrodynamic size of particles, and the effectiveness of surface functionalization were monitored with the help of the DLS technique. Following the functionalization, the size and shape of the metallic core were observed using the HR-STEM technique.

### 2.2. Synthesis Poly(dimethylsiloxane) Cross-Linked by Metal-Ligand Coordination with the Addition of AuNPs

We used PDMS cross-linked through metal–bpy coordination by zinc chloride salt (bpyPDMS-ZnCl2). A modified synthetic path was used according to Williams et al. [28]. The amide condensation reaction (under nitrogen atmosphere) was performed by the slow addition of aminopropyl terminated PDMS (Mn = 3350 g/mol, Gelest Inc.) to a previously cooled (273 K) mixture of 2,2′-bipyridine-4,4′-dicarboxylic acid (0.100 g, 1.2 eq.), 4-dimethylaminopyridine (0.007 g, 0.2 eq.), and N-(3-dimethylaminopropyl)-N′-ethylcarbodiimide hydrochloride (EDCI, 0.170 g, 2.5 eq.) dissolved in dry methylene chloride DCM (50 mL). The reaction was left for 15 h at room temperature (~293 K). Product purification was executed by first washing twice with 0.5 M NaOHaq (50 mL) in order to remove an excess of 2,2′-bipyridine-4,4′-dicarboxylic acid, and then once with 0.5 M HClaq (50 mL) in order to eliminate side products of the EDCI compound. Finally, the material obtained was washed with saturated NaHCO_3_aq (50 mL) and saturated NaClaq (50 mL). The organic phase was then concentrated using a rotary evaporator to a minimum DCM (~0.5 mL) and crashed out upon adding methanol (2 mL). The resulting yield amounted to 55% (0.6 g, purity ~90%), with the 1H NMR spectrum being presented in our previous work [22]. To obtain bpyPDMS-ZnCl2, 0.1 g (2.8 × 10^−5^ M), bpyPDMS was dissolved in 10 mL of toluene and left for 12 h. Then, 30 μL (9.3 × 10^−6^ M) aliquot of a methanol solution containing ZnCl_2_ salt (Tokyo Chemical Industry, Tokyo, Japan) was added to the bpyPDMS solution in order to obtain PDMS cross-linking by metal-ligand coordination, with the metal: ligand coordination stoichiometry amounting to 1:3 [29]. To the prepared organometallic compound, gold nanoparticles (AuNPs) were added in the amount of: 0.001 wt. %, 0.0025 wt. %, 0.005 wt. %, 0.04 wt. %, 0.2 wt. %, 0.4 wt. %, functionalized with PSSH to ensure the stability and dispersibility of nanoparticles in the PDMS matrix [27].

bpyPDMS solution to obtain PDMS cross-linking by metal-ligand coordination with 1:3 metal: ligand coordination stoichiometry [28]. For such prepared organometallic compound, we added gold nanoparticles (AuNPs) in the amount of: 0.001 wt. %, 0.0025 wt. %, 0.005 wt. %, 0.04 wt. %, 0.2 wt. %, and 0.4 wt. % of AuNPs, which were functionalized with PSSH to ensure the stability and dispersibility of nanoparticles in the PDMS matrix [27].

## 3. Techniques

### 3.1. Transmission Electron Microscopy (TEM)

Characterization of the size and shape of the gold nanoparticles before and after functionalization and the AuNPs-PSSH distribution in the PDMS matrix was performed by means of a high-resolution scanning electron microscope (HR-SEM) equipped with a STEM II detector (NovaNanoSEM 450, FEI, USA; immersion mode; accelerated voltage = 30 kV; spot size=1.5; images acquired at the bright field (BF), darkfield (DF) detector modes); with the specimens for measurements being prepared by drop-casting of the colloid on carbon-coated copper grids.

### 3.2. Dynamic Light Scattering (DLS)

Dynamic sight-scattering (DLS) measurements (colloidal stability and hydrodynamic size of AuNPs before and after functionalization) were performed using DLS, Litesizer 500, Particle Analyzer, Anton Paar (at room temperature, in a quartz cuvette).

### 3.3. Broadband Dielectric Spectroscopy (BDS)

BDS measurements within a frequency range from 1 Hz to 1 MHz and temperature range from 143 K to 373 K were performed using a Novocontrol^®^ GmbH Concept 80 Broadband Dielectric Spectrometer (Montabaur, Germany) equipped with a Quatro Cryosystem (Montabaur, Germany) providing stability better than 0.5 K. For the measurements, high-quality Novocontrol interdigitated electrodes with diameter 15 mm, electrode basic structure of size 75 μm, and loss factor tan(δ) ~0.001 were used. The materials investigated were dissolved in toluene and poured out on the interdigitated electrodes. Since several layers were required in order to fill out the space between the interdigitated electrodes, after applying each layer, the solvent was allowed to evaporate on the hot plate. Before carrying out dielectric measurements, the samples were kept overnight in a vacuum dryer at elevated temperatures.

Dielectric properties [30] of the investigated materials are represented as isothermal plots of:-Relative permittivity (ε′) that is part of complex dielectric function ε* defined by:
ε* (ω) = ε′ (ω) − iε″ (ω)(1)
where ε ′(ω) is the real part and ε ″(ω) the imaginary part of the complex dielectric function, i = $\sqrt{-1}$ and ω denotes angular frequency.-Tangent delta called loss tangent, which is described as tangents of the phase angle between ε′(ω) and ε″(ω)
(2)$$\tan\sigma =\frac{\varepsilon^{\prime \prime }}{\varepsilon^{\prime }}$$-Real part of conductivity (σ′)
(3)$$\sigma^{\prime }\left(\omega \right)={\omega \varepsilon }_{0}\varepsilon^{\prime \prime }\left(\omega \right)$$
where $\varepsilon_{0}$ is the dielectric permittivity of vacuum ($\varepsilon_{0}$ = 8.854 × 10^−12^ As·V^−1^·m^−1^).-Imaginary part of modulus (M″)
(4)$$M^{\prime \prime }\left(\omega \right)=\frac{\varepsilon^{\prime \prime }\left(\omega \right)}{\varepsilon^{\prime 2}\left(\omega \right)+\varepsilon^{\prime \prime 2}\left(\omega \right)}$$-Additionally, Arrhenius plots of conductivity relaxation times (τ_c_) can be defined as follows:
(5)$$\tau_{C}\left(T\right)=\tau_{0}\exp\left(\frac{-E_{A}}{k_{B}T}\right)$$
where τ_0_ denotes pro-exponential factor, E_A_ denotes activation energy, k_B_ denotes Boltzmann constant, and T denotes absolute temperature.

Equations (1) and (2) were determined at the temperature of 313 K, Equation (3) at that of 368 K, and Equation (4) in the temperature range of 143–373 K. All the plots cover a broad frequency range of 1 Hz–1 MHz.

## 4. Results and Discussion

### 4.1. AuNPs Characterization by DLS and HR-STEM

DLS and HR-STEM measurements were performed in order to determine the colloidal stability and to characterize the size, shape, and size distribution of AuNPs in water (see Figure 1), as well as to determine and confirm the strength of the nanoparticles after their PSSH functionalization via phase transfer process (see Figure 2). DLS measurements revealed the colloidal stability of AuNPs synthesized in water with their hydrodynamic size equal to d_H-DLS_ = 9 ± 2 nm (Figure 1A). The HR-SEM images confirmed the spherical shape of particles with the size of metallic core equal to d_STEM_ = 5 ± 1 nm (Figure 1B,C).

The colloidal stability of AuNPs after functionalization with PSSH ligand was monitored with DLS and HR-STEM measurements (see Figure 2). DLS measurements revealed an increase in the hydrodynamic diameter of nanoparticles after their functionalization from d_H-AuNPs_ 9 ± 2 nm (AuNPs in water) to d_H-AuNPs-PSSH_ = 37 ± 13 nm (PSSH-functionalized AuNPs in toluene). This increase in the hydrodynamic size of the nanoparticles is related to the modification of their surface with a large PSSH polymer ligand. DLS measurements confirmed the colloidal stability of PSSH-functionalized AuNPs after the phase transfer process—no aggregates or agglomerates were detected in the system. An analysis of HR-STEM images of PSSH-functionalized AuNPs (Figure 2B–D) confirmed the stability of the nanoparticle size and shape after the functionalization process—the size of the metallic core remains unchanged and amounts to d_STEM_ = 5 ± 1 nm. However, the analysis of particle distribution deposited on carbon-coated copper grids for the HR-STEM measurements revealed a change in the functionalized nanoparticle arrangement on a solid substrate (Figure 2B–D) compared to that of the non-functionalized AuNPs (Figure 1B). The presence of PSSH ligands on AuNPs brings about separation and homogeneous distribution of the nanoparticles both in the colloid state (which is confirmed by the hydrodynamic size of the particles) and on the solid substrate after solvent evaporation (which is confirmed by the arrangement of particles on carbon-coated copper grids). An analysis of mutual distances of the PSSH-functionalized AuNPs performed on the basis of the HR-STEM image (Figure 2C) has shown these distances to amount to approximately 30 nm. This remains in suitable agreement with the hydrodynamic size of the nanoparticles after their PSSH functionalization, which is also about 30 nm.

### 4.2. Broadband Dielectric Spectroscopy

#### 4.2.1. Isothermal Representation of Dielectric Properties of the Systems Investigated

The results of the real part of dielectric permittivity (ε′) determination of PDMS and metalloorganic complexes with various wt. % of AuNPs measured at 313 K in the frequency range 10^2^–10^6^ Hz are depicted in Figure 3. One can see that (with the exemption of the bpyPDMS-ZnCl_2_ system where a slight increase is observed at low frequencies), the values of ε′ do not change over a wide frequency range for all the materials investigated, which agrees well with our previous findings [22,23]. An approximately two times higher value of ε′ for bpyPDMS-ZnCl_2_, compared to the neat PDMS matrix (ε′ = 2.4, which remains in a suitable agreement with the literature data [30]), results from an introduction of additional dipoles into the system and a formation of bipolar metal-ligand coordination bonds following the cross-linking process [31,32]. Adding AuNPs to the system brings about a slight decrease in dielectric permittivity. For the nanoparticle content between 0.001 and 0.2 wt. %, the ε′ value amounts to approximately 4.5. However, in the case of the system comprising 0.4 wt. % of AuNPs, its magnitude is lower, and it remains below 4. That small decrease in dielectric permittivity of the systems containing gold nanoparticles can be explained by the large specific surface area of AuNPs, bringing about stronger interaction with active dipoles of the polymer matrix and thus reducing its polarization. A similar effect was found by Wang et al., who observed a small decrease in dielectric permittivity in epoxy resin composites with ZnO nanoparticles [33]. The authors explained the effect as being a result of an interaction with the polymer matrix, restricting chain movement and thus reducing both space charge and free volume of the nanocomposites. As evident from Figure 3 and Figure 4, a similar decrease in dielectric permittivity and loss tangent is observed in the present work at high frequencies.

The values of loss tangent (loss tan), determined at 313 K within a broad range of frequencies for neat PDMS and its metalloorganic complexes with various contents of AuNPs, are presented in Figure 4. Neat PDMS is characterized by the lowest value of loss tan in the frequency range of 10^3^ ÷ 10^6^ Hz among all the samples investigated. In the lower frequency range, the absorption response grows along with decreasing frequency due to an increasing contribution of direct conductivity effects. In the case of PDMS cross-linked by metal-ligand coordination, an increase in loss tangent values has been observed in the frequency range of 10^3^ ÷ 10^6^ Hz. This finding had been expected to be directly connected with an intentional introduction of mobile Cl^−^ counter anions originating from ZnCl_2_ salt that was used in the cross-linking procedure. On the other hand, in the frequency range below 3 × 10^2^ Hz, loss tangent values for the cross-linked specimens are lower than for the neat PDMS. At the temperature of 313 K, neat PDMS is characterized by low viscosity, whereas its complexes with AuNPs exhibit a solid-state nature. This difference results in a change of conduction mechanism, which will be discussed later in this work and which directly influences loss tangent. It is evident from Figure 4 that the loss tangent is greatly affected by adding AuNPs into the systems investigated. It should be noted that an addition of 0.001 wt.% of AuNPs only results in the decrease in the tangent loss in the entire frequency range. A slight difference in the loss tangent value in the MHz frequency range, between 0.007 and 0.004 for bpyPDMS-ZnCl_2_ and the systems containing 0.4 wt.% AuNPs, reflect less energy dissipation due to restricted dipole mobility. This remains in suitable agreement with the observed decrease in dielectric permittivity values, presented in Figure 3. The more pronounced effect of AuNPs on the loss tangent value can be observed in a low-frequency region, which is more sensitive to conductivity. The loss tangent value at 1.15 Hz varies depending on the content of AuNPs. In the case of bpyPDMS-ZnCl_2,_ loss tangent value amounts to 0.38, whereas for its complexes with AuNPs at concentrations between 0.001 wt.% and 0.04 wt.%, it oscillates around 0.02.

While increasing the nanoparticle concentration to 0.4 wt.% results in the rise of loss tangent value up to 0.05, further increase in that concentration, namely above 2 wt.%, leads to a shortcut circuit (data not shown).

#### 4.2.2. Electrical Conductivity of Neat PDMS, Its Metalloorganic Complex bpyPDMS-ZnCl_2_, and the Materials Containing Gold Nanoparticles

Figure 5a,b presents the real part of conductivity (σ′) as a function of frequency for PDMS and for all the organometallic compounds with various contents of AuNPs investigated. It is well documented in the literature [34,35] that the dispersion of AC conductivity for nonhomogeneous as well as disordered solids increases with increasing frequency. Interesting and still not fully understood is the fact that the AC conductivity of various types of disordered solids exhibits a similar type of frequency dependence [34,36,37]. On the other hand, their DC conductivity is closely related to the concentration and the mobility of charge carriers. Consequently, its contribution is both facilitated and reinforced by cooperative movements of the units or chains of the polymer molecule, typical for either viscoelastic or molten polymer state.

In Figure 5, a plateau is observed in the low-frequency region (σ_dc_). This suggests that the conduction is solely due to the migration of electric charges, thus acquiring a certain constant/plateau value. A careful analysis of the data presented in Figure 5 allows one to predict the frequency point at which DC conductivity contribution begins to overrule the AC’s component [38]. In the high-frequency range investigated, dipole relaxations predominate the dielectric response of the material. The beginning of the frequency-dependent conductivity corresponds to the characteristic time τ_c_:(6)$$\tau_{c}=\frac{1}{2{\pi \omega }_{c}}$$
where ω_c_ denotes crossover frequency [39], and it can be assigned to a maximum value of M″ vs. frequency representation (see Figure 7). According to the theory of linear conductivity response, (τ) >, assuming that the movement of ions in thermal equilibrium is diffusive and its mean square shift <λ^2^ (t)> increases linearly with time. Above the crossover frequency, i.e., when ω > ω_c_, σ′(ω) representation increases with frequency, and (τ) <. In this case, the movement of ions is governed by a sub-diffusive mechanism; however, their mean square shift also increases linearly with time, as in the case of diffusive ions. It should also be noted that the initial frequency for the dispersion part of the conductivity is thermally activated, with the activation energy being identical to the activation energy of DC conductivity σ_dc_ [39].

In disordered materials, one can observe universalities in the AC conductivity, which are clearly visible in the σ′ vs. frequency graph (Figure 5). In the particular case when ω > ω_c_, a gradual dispersion of AC conductivity takes place, thus resulting in the following power law dependence:(7)$$\sigma \left(\omega \right)\sim \omega^{S}$$
where the s parameter remains in the range 0.5 ≤ s ≤ 1, and it increases with decreasing temperature and increasing frequency.

In the data presented, the conductivity is dependent on frequency over a nearly entire frequency range, following a universal law expressed by Equation (7) with the exponent s being close to 0.9 at 363 K. It is only at low frequencies that a certain deviation from linearity is observed, especially for neat PDMS. In this case, within the range below ca. 10^2^ Hz, the conductivity is practically independent of frequency, and it amounts to 1.5 × 10^−12^ S/cm at 323 K, which remains in suitable agreement with the literature data [40]. As far as the PDMS sample behavior in the high-frequency region is concerned (see Figure 5), an AC conductivity contribution is observed, which can be assigned to rapid movements of charge carriers at small distances inside the Coulombic cage. In contrast, in the low-frequency range, primarily DC conductivity contribution is observed in the form of a plateau (not visible in Figure 5 for investigated systems with AuNPs). This result can be associated with the fact that ions/charge carriers have to jump over longer distances and overcome larger energy barriers. It can also be understood as defining conditions for the ions leaving the Coulombic cage. In the case of the compound cross-linked by metal-ligand coordination (bpyPDMS-ZnCl_2_), the Cl^−^ counter anions assume positions near zinc cations due to Coulombic interactions between each other. When measured at 313 K, all the materials synthesized exhibit smaller conductivity in the low-frequency range (in comparison to neat PDMS) because they remain in a solid state under these conditions, while PDMS is a viscoelastic material of low viscosity. A detailed analysis of the data presented in Figure 5 provides valuable information concerning the effect of the addition of gold nanoparticles on the PDMS metalloorganic complex. Following the addition of 0.001 wt. % of AuNPs into the bpyPMDS-ZnCl_2_ system, the σ′ value decreases approximately one order of magnitude in the low-frequency range compared to that of bpyPDMS-ZnCl_2_ (see Figure 5b). It can also be recognized that the addition into the systems investigated of AuNPs at concentrations up to 0.04% wt. did not result in any change of the σ′ value. Above this concentration, gold nanoparticles exhibit weaker trapping ability, thus enforcing the DC component by their very presence and movement in the electric field. It can be seen in the inset of Figure 5, where the σ′ values increase in the low-frequency region for complexes with the content of 0.2 and 0.4% wt. AuNPs.

An evident change of the σ′ values at low frequencies with the smallest and short DC conductivity plateau is observed for samples with an addition of AuNPs. The results confirm the effect of AuNPs on the electrical properties of the materials investigated. These phenomena can be explained by trapping Cl^−^ counter ions by the active surface of AuNPs [41].

Large active surface (see Figure 6) of AuNPs, comprised in the bpyPDMS-ZnCl_2_ complexes, affects the Cl^−^ counterions by attraction, thus resulting in the restriction of these charge carriers’ mobility and in the decrease in the conductivity values in the low-frequency region compared to those of bpyPDMS-ZnCl_2_ without nanoparticles.

In the frequency range above ω_c_, Coulombic forces between AuNPs and Cl- ions are getting too weak to keep the ions close to the nanoparticle surface and, therefore, to confine them inside the Coulombic cage. For frequencies above ω_c_, several microscopic models have been introduced in order to describe conductivity. Currently, the most popular way to describe the conduction mechanism in this range is that of charge hopping, and numerous models have been developed over the years. The one widely used is the random free-energy barrier model proposed by Dyre [37]. This particular approximation assumes that conduction occurs by means of hopping of charge carriers subjected to randomly varying spatial energy barriers. That model appears to be the most suitable for the systems investigated in this work. However, there are other models approximating conductivity mechanisms in disordered materials present in the literature. The simplest macroscopic model of electrical conduction is based on percolation [42,43]. Another model considered the sample to be a mixture of components having different dielectric properties with averaged effective conductivity being calculated in a self-consistent manner [29,44]. It is also possible to model the studied materials by equivalent electrical circuits of random resistors and capacitors [45].

#### 4.2.3. DC and AC Electrical Conductivity Contribution to System Complex Conductivity

Frequency spectra of the imaginary part of modulus M″(ω) (loss modulus) at 368 K for PDMS and all the investigated organometallic compounds with various wt. % of AuNPs are presented in Figure 7. A temperature of 368 K was selected because only at such a high temperature the contribution of the DC component is observed for all the samples.

An analysis of the maxima presented in Figure 7 is particularly meaningful. Each of the maxima of M″(ω) representation recorded reveals a critical point where ion movement changes from diffusive to sub-diffusive. Additionally, this particular point can be related to the moment when ions begin to leave the Coulombic cage. The differences between the M″(ω) maxima detected for bpyPDMS-ZnCl_2_ and for neat PDMS can be connected with the intentional introduction of ZnCl_2_ used in the cross-linking procedure [23]. An evident maximum of M″(ω) representation is observed at a similar frequency around 10 Hz at 368 K. Introducing 0.001 and 0.04 wt. % of gold nanoparticles into bpyPDMS-ZnCl_2_ brings about a shift of the imaginary part of the modulus M″(ω) maximum toward lower frequencies, i.e., toward higher temperatures in the temperature representation. Further addition of larger amounts of nanoparticles results in a shift of maximum toward higher frequencies to the position of PDMS. It should be pointed out that the system synthesized comprising metallic nanoparticles exhibits fully capacitive characteristics with the phase angle being close to 90°. The content of AuNPs higher than 2 wt. % rings about a short circuit.

A modification of the conduction mechanism, within the meaning of the change of place where the DC conductivity component begins to be dominant, can be directly linked to the molecular structure of the system investigated. In the case of bpyPMMA-ZnCl_2_, the Cl^−^ counter anions are placed in the neighborhood of zinc. In the case of samples with the addition of AuNPs, counter anions are “trapped” by the active nanoparticle surface.

The maxima of M″(ω) representation constitute the points where DC conductivity contribution begins to dominate the conduction mechanism for each sample (see Figure 7). These data are essential for the construction of a conductivity relaxation map for bpyPDMS-ZnCl_2_ and for all the investigated metalloorganic complexes with various wt. % of AuNPs as a function of 1000/T (see Figure 8). Based on the maximum characteristic frequency plot in M″ (that is ω_σ_), conductivity relaxation time (τ_c_) was calculated with the help of the following equation: τ_c_ = 1/(2πω_σ_). Such data obtained in a broad range of temperatures provide a very important tool used to distinguish between different charge carrier transport mechanisms in the systems investigated. It can be observed that the temperature at which the DC component begins to contribute depends on wt. % addition of AuNPs.

From the application point of view, the most preferable situation is when DC conductivity demonstrates the smallest possible value and begins to dominate at the highest possible temperature. As far as samples containing gold nanoparticles are concerned, the DC conductivity contribution occurs at higher temperatures than it is in the case of bpyPDMS-ZnCl_2_. The subsequent activation of the conduction mechanism, taking place at higher temperatures, can potentially generate a lower leakage current in organic electronic devices. It can be concluded that the most effective addition wt. % of AuNPs remains in the range of 0.001 wt. % to 0.04 wt. %.

To summarize, the conductivity depends on two factors—the density of the charge carriers and their mobility. On the assumption that AuNPs work as effective Cl^−^ counter anions traps, one can state that the conductivity in the polymer matrix is controlled by their concentration. In the sample of bpyPDMS-ZnCl_2_ free of AuNPs, the chloride counter ions are released much easier from the Coulomb cage as the Coulombic interaction between zinc and Cl^−^ is substantially weaker than that between nanoparticles and Cl^−^.

On the basis of the maximum characteristic frequency M″ map for PDMS and all the investigated metalloorganic complexes with various wt. % of AuNPs as functions of 1000/T within the temperature-frequency range investigated, the conduction mechanism can be interpreted by the Arrhenius approach. The temperature at which DC contribution becomes a predominant factor in the conduction mechanism significantly increases for the samples containing AuNPs compared to those of neat PDMS and bpyPDMS-ZnCl_2_. It is essential to underline that even a very small amount of AuNPs acts as an effective trap system damping conductivity by decreasing ion mobility in the bpyPDMS-ZnCl_2_ coordination complexes within the low-temperature range investigated. This effect arrives at its optimum at the nanoparticle content from 0.001 to 0.04 wt. %. For higher concentrations of gold nanoparticles, i.e., between 0.2 and 0.4 wt.%, the DC values become higher, but they are still lower than in the case of the Zn-bpy complex. Under these conditions, AuNPs still fulfill their role as traps by lowering the mobility of chloride ions. Our findings remain in suitable agreement with those of previously published works reporting non-volatile memory devices where the significant charge trapping ability of gold nanoparticles was confirmed [41].

## 5. Conclusions

In the present work, gold nanoparticles (5 ± 1 nm) were synthesized in an aqueous environment by means of a chemical reduction method and later functionalized by thiol-terminated polystyrene (PSSH) (37 ± 13 nm). As the next step, 0.001 wt. %, 0.0025 wt. %, 0.005 wt. %, 0.04 wt. %, 0.2 wt. %, 0.4 wt. % of stabilized AuNPs were added to poly(dimethylsiloxane) (PDMS) cross-linked by ZnCl_2_-bipyridine coordination. Broadband dielectric spectroscopy was employed in order to investigate the effect of AuNPs addition on the dielectric and electrical properties of so synthesized composites. It has been found that the conductivity of the materials studied decreases upon adding AuNPs, and it is damped in the low-temperature range. An introduction of 0.001 wt. % AuNPs into polymeric matrix already reduces the values of conductivity and loss tan by nearly one order of magnitude at 1 Hz, whereas the real part of dielectric permittivity is only marginally lowered. The conductivity activation map demonstrates that DC contribution of conductivity for samples containing AuNPs is observed at significantly higher temperatures than in the case of both neat PDMS and bpyPDMS-ZnCl_2_. Addition of as little as 0.001 wt. % of the AuNPs decreases ion mobility and damping conductivity below 363 K. Our study shows that the concentration of AuNPs in bpyPDMS-ZnCl_2_ up to 0.04 wt. % makes optimum conditions. Larger amounts of gold nanoparticles make the DC conductivity values begin to increase (while still being lower than in the bpyPDMS-ZnCl_2_ system investigated). The lower conductivity of materials with gold nanoparticles can be explained by the trapping mechanism and the lowering of the mobility of chloride ions caused by the active surface of AuNPs. BpyPDMS-ZnCl_2_ system with 0.04 wt. % of AuNPs exhibits an optimum dielectric permittivity value and lower DC conductivity. For that very reason, the systems referred to seem to be the most promising candidates for elastic dielectrics in the OFET technology, where low values of gate leakage current are required.

## Figures and Tables

**Figure 1 molecules-27-03579-f001:**
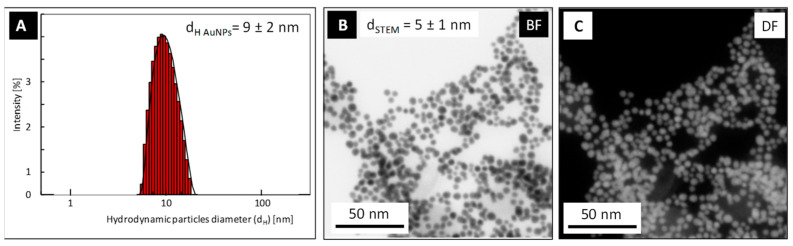
DLS size distribution histogram (**A**) and HR-STEM images (bright field—(**B**); dark field—(**C**)) of aqueous AuNPs.

**Figure 2 molecules-27-03579-f002:**
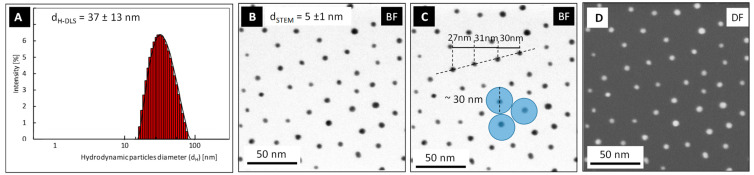
DLS size distribution histogram (**A**) and HR-STEM images (bright field—(**B**,**C**); dark field—(**D**)) of PSSH-functionalized AuNPs.

**Figure 3 molecules-27-03579-f003:**
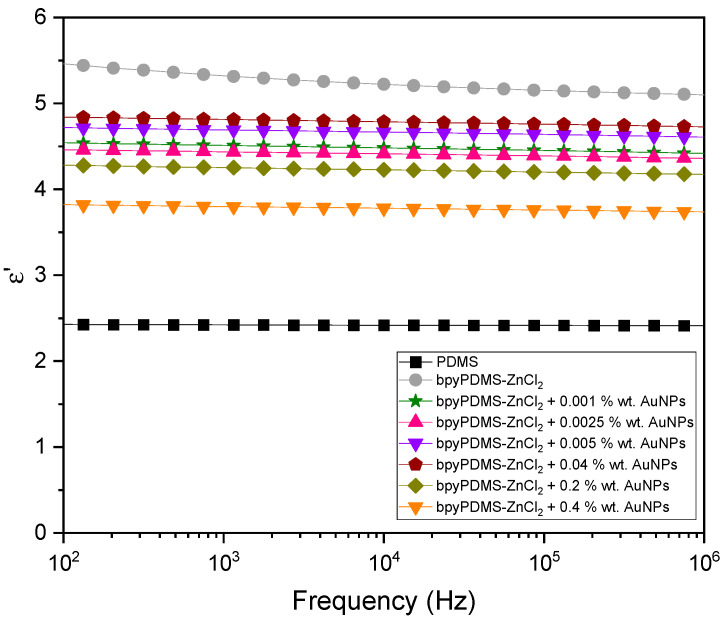
Frequency dependence of the real part of dielectric permittivity (ε′) at 313 K of PDMS and metalloorganic complexes with various contents of AuNPs.

**Figure 4 molecules-27-03579-f004:**
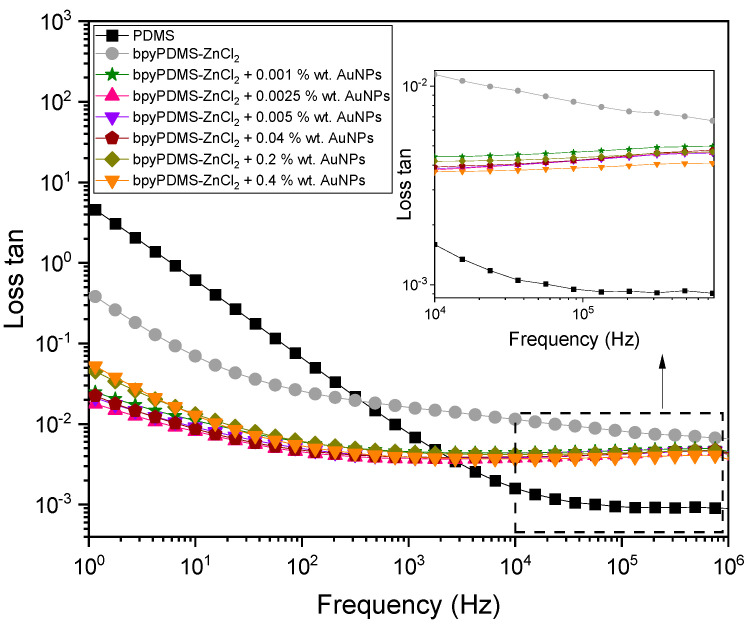
Frequency dependence of loss tangent determined at 313 K of neat PDMS and its metalloorganic complexes with various contents of AuNPs.

**Figure 5 molecules-27-03579-f005:**
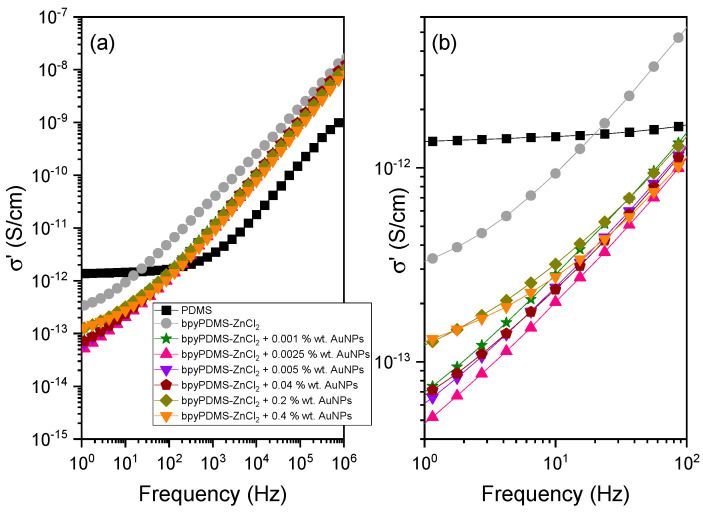
(**a**) The real part of conductivity (σ′) of neat PDMS and all the organometallic compounds with various content of AuNPs as frequency function recorded at 313 K. (**b**) shows the enlargement of these representations in the low-frequency region.

**Figure 6 molecules-27-03579-f006:**
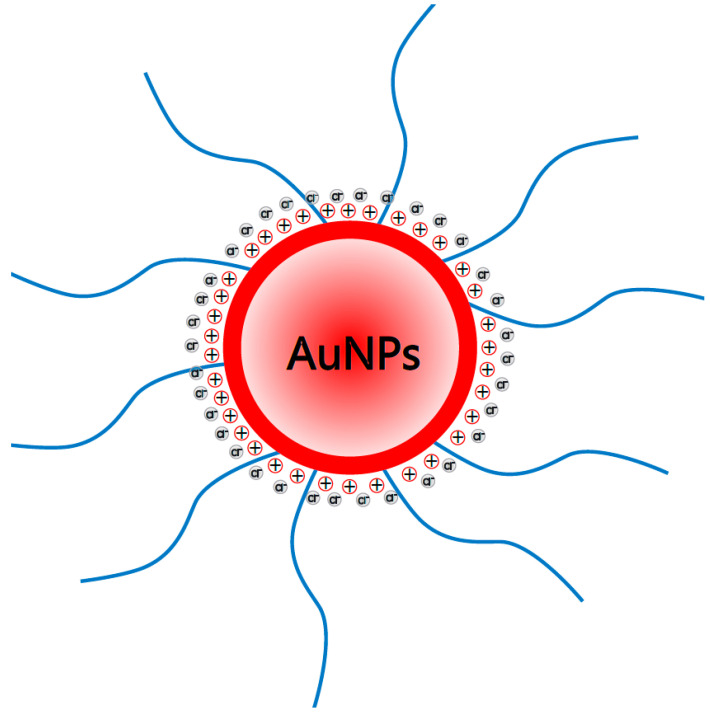
Schematic view of trapping Cl^−^ counterions by gold nanoparticles (AuNPs) in poly(dimethylsiloxane) cross-linked with metal-ligand coordination (bpyPDMS-ZnCl_2_ + AuNPs), where positive charge remains on the nanoparticle surface, and blue lines represent PSSH polymer.

**Figure 7 molecules-27-03579-f007:**
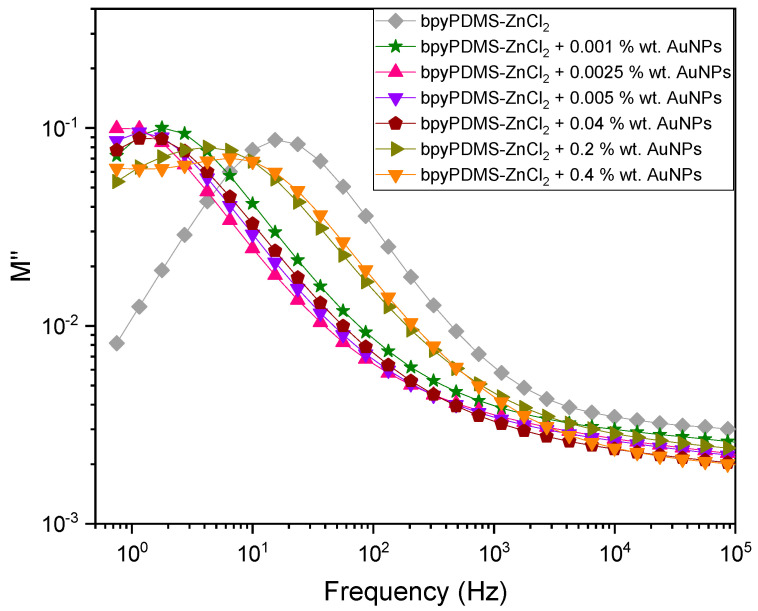
The imaginary part of modulus (M″) for PDMS and all the investigated organometallic compounds with various wt. % of AuNPs vs. frequency at 368 K.

**Figure 8 molecules-27-03579-f008:**
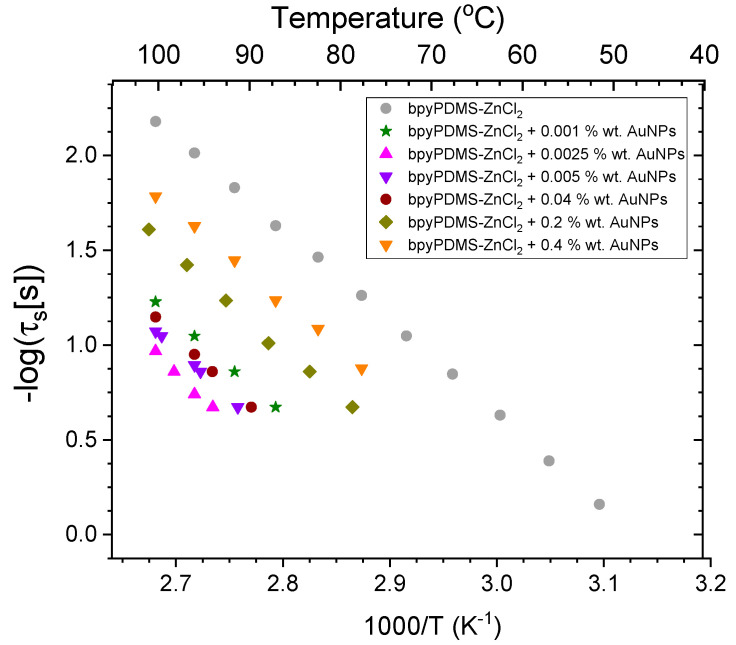
Conductivity activation map for bpyPDMS-ZnCl_2_ and for all the investigated metalloorganic complexes with various wt. % of AuNPs.

## Data Availability

Not applicable.

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
