# Peer review of "Gold Nanoparticles as Effective ion Traps in Poly(dimethylsiloxane) Cross-Linked by Metal-Ligand Coordination"

_molecules, 2022, doi:10.3390/molecules27113579_

Round 1

Reviewer 1 Report

The authors prepared a AuNPs dopping elastic dielectrics with PDMS as the matrix which has the potential application in OFETs. The manuscript was nicely organized and should be published in Molecules after some minor revisions.

1.  The sentence of "DLS and HR-STEM measurements were performed to determine the colloidal stability and to characterize the size, shape, and size distribution of AuNPs in water (Figure 1), as well as to determine and confirm the strength of AuNPs after PSSH functionalization via phase transfer process (Figure 2)." in L. 208, Pg. 5 seems not necessry to section 4.1.

2. "The differences between detected peak maximum of the imaginary part of modulus M”(ω) in sample cross-linked by metal-ligand coordination in comparison to neat PDMS can be connected with intention introduction of metal salt (ZnCl2) used for cross-linking procedure" in L.403, Pg. 11 needs rewritten in more concise way.

3. The axis titles of  Fig. 1A, 2A and the inset of Fig. 4 should be enlarged to a more readible size. 

4. The size of 'Cl-'  in Fig. 6 is too small to easily read.

Author Response

We would like to thank the reviewer for the valuable comments. We have made changes to improve the quality and clarity of our work and thoroughly revised the manuscript. The changes made are marked in yellow in the revised version of the manuscript. We also made English corrections as have been suggested.

We have replied to all the comments, point by point, as listed below.

  1. We would like to keep this sentence in its present form as introduces the reader to the next part of the text.
  2. The sentence has been rewritten as follows:

The differences between the M”(ω) maxima detected for bpyPDMS-ZnCl2 and for neat PDMS can be connected with the intentional introduction of ZnCl2 used in the cross-linking procedure[23].

  1. The axis titles of  Fig. 1A, 2A, and the inset of Fig. 4 have been enlarged to a more readable size. 
  2. The size of 'Cl-'  in Fig. 6 was made on this size in purpose. In the present form, the sizes reflect the proportions between AuNPs and Cl_.

Reviewer 2 Report

The present work is a continuation of previous studies of the group. The research of PDMS with AuNPs composites is comprehensive, the text is written clearly and unambiguously, the research methods and interpretation of the results do not raise doubts. However, minor revisions should be made to the manuscript.

  1. Are AuNPs particles metallic inclusions in organics or positively charged? Gold complexes H[AuCl4] decompose from 150C to AuCl, which is insoluble and yellow in color. Does the coloration of the composite and its intensity change with changes in AuNPs concentration?
  2. t is necessary to add a comparison of conductivity and dielectric constant with values from the literature because the basic polymer material has been widely studied before. For example: Xiaolong Gao and etc Mechanically Enhanced Electrical Conductivity of Polydimethylsiloxane-Based Composites by a Hot Embossing Process. Polymers (Basel). 2019 Jan; 11(1): 56. doi: 10.3390/polym11010056
  3. Is the conductivity of composites with 0.001-0.04% additive the same ( Fig. 5 right ), taking into account the error in determining the values ?
  4. The list of references should be corrected according to the requirements of the journal, in particular references 4, 7, 17, and 19.
